# Cross-sectional study of asymptomatic Neisseria gonorrhoeae and Chlamydia trachomatis infections in sexually transmitted disease related clinics in Shenzhen, China

**Shu-Xia Chang**[1☉], **Kang-Kang Chen**[2☉], **Xiao-Ting Liu**[3], **Nan Xia**[4], **Pei-Sheng Xiong**[2], **Yu-Mao Cai**[5]*

1 Shenzhen Longgang Center for Chronic Disease Control, Shenzhen, China, 2 Department of Preventive Medicine, Shantou University Medical College, Shantou, China, 3 Department of Nursing, Shantou University Medical College, Shantou, China, 4 Department of Preventive Medicine, Central South University, Changsha, China, 5 Shenzhen Center for Chronic Disease Control, Shenzhen, China

☉ These authors contributed equally to this work.
* caiyumao@163.com

**Data Availability Statement:** All relevant data are within the manuscript and Supporting Information files.

## Abstract

The aims of this study were to investigate the prevalence and proportion of laboratory-confirmed urethral *Chlamydia trachomatis* (CT) and *Neisseria gonorrhoeae* (NG) infections that were asymptomatic among individuals presenting to clinics in Shenzhen and the risk factors related to asymptomatic CT infection. In a cross-sectional study, eligible individuals were invited to participate in the questionnaire, and urine specimens were collected to identify CT and NG infections using a nucleic acid amplification test (NAAT). Considering the differences in the presentation of symptoms between men and women, this study was stratified by gender. Corresponding outcomes were analyzed by Chi-square test and multivariate logistic regression. A total of 2,871 participants were asymptomatic and included in our analyses: 1120 (39.0%) men and 1751 (61.0%) women. The prevalence of asymptomatic NG and CT infections was 0.9% and 6.2% in men, and 0.4% and 7.9% in women, respectively. The proportion of asymptomatic urethral CT among men with urethral CT was 28.3%; for women, it was 34.2%. For asymptomatic men with CT, 3 independent risk factors were identified: (1) men under the age of 30 (aOR, 1.83; 95% CI, 1.11–3.03); (2) being employed in the commercial service work (2.82; 1.36–5.84); and (3) being recruited through the urological department (2.12; 1.19–3.79). For asymptomatic women with urethral CT, age less than 30 years was a risk factor. In conclusion, a substantial prevalence of asymptomatic CT infections was found among men and women presenting to clinics in Shenzhen. The significant correlation between asymptomatic CT infection and these risk factors could help identify high-risk populations and guide screening.

**Funding:** This work was supported by [Sanming Project of Medicine in Shenzhen] grant number [SZSM201611077] to SC.

**Competing interests:** The authors declare that they have no conflict of interest.

**Abbreviations:** (CT), Chlamydia trachomatis; (NG), Neisseria gonorrhoeae; (STIs), sexually transmitted infections; (CI), Confidence interval; (AORs), Adjusted odds ratios.

## Introduction

*Chlamydia trachomatis* (CT) and *Neisseria gonorrhoeae* (NG) are the first and second most common bacterial sexually transmitted infections (STIs), with a global incidence of respectively 127.2 million and 86.9 million in 2016.[1] CT and NG usually colonize and infect the human reproductive tract; if left untreated or improperly treated, they can lead to severe complications, such as penile stricture and epididymitis in men, pelvic inflammatory disease and endometritis in women, and eventually lead to infertility in both genders.[2] Furthermore, NG and CT infections are also risk factors associated with the transmission and infection of HIV.[3] In the US, medical costs for gonorrhea and chlamydia are estimated at $162.1 and $516.7 million, respectively.[4]

However, due to differences in the distribution of risk factors, the prevalence and burden of STIs vary widely around the world. Shenzhen is a newly developed city with the floating population accounting for about 87% of the total population,[5] which makes it very likely to be a hotbed for STIs. The latest molecular epidemiological study on genital NG and CT infections conducted by Zhang et al[6] in Shenzhen in 2009 showed that the prevalence of CT and NG among participants presenting to clinics was 17.7% and 9.7%, respectively. The prevalence of CT and NG observed in this study was considerably higher than the results in the Chinese Health and Family Life Survey where the overall prevalence of CT and NG infections was 2.6% and 0.08% in women, and 2.1% and 0.02% in men, respectively.[7] Therefore, Shenzhen may be an appropriate place to study risk factors associated with STIs, and evidence-based interventions to reduce the burden of STIs in Shenzhen may be more cost-effective. However, Zhang and colleagues' study lacks important information, such as the prevalence and proportion of asymptomatic cases and risk factors for asymptomatic infections. Asymptomatic infections will undoubtedly further facilitate the spread of gonorrhea and chlamydia, because people with gonorrhea/chlamydia but no symptoms are less likely to seek any treatment. In addition, knowledge of the burden and risk factors of asymptomatic STIs may have implications for syndromic management which is the primary care for the detection and treatment of suspected STI infections in resource-limited settings.[8, 9] At present, the effectiveness of syndromic management on reduction of the prevalence of STI infections is not satisfactory.[10–12] The reason is not only because of its poor sensitivity, but more importantly, most STI infections such as CT and NG are asymptomatic.

Here we investigate the prevalence and proportion of laboratory-confirmed urethral CT and/or NG infections that were asymptomatic among individuals presenting to clinics in Shenzhen and the risk factors related to asymptomatic CT infection. Given the limited health resources, there is currently no guidelines for chlamydia and gonorrhea screening in China. The findings from this study may help us to ensure proper resource allocation and develop intervention activities.

## Materials and methods

### Sampling methods and recruitment

Participants in our study were recruited from 1 April to 16 May 2018 by using the stratified purposive sampling method. First, based on the number of NG and CT cases reported in Shenzhen in 2017, we selected the 6 administrative districts with the largest number of reported cases from the 10 administrative districts in Shenzhen. Then, in each of the selected districts, four hospitals with a high number of reported cases were included, except 1 district with only 2 hospitals. Finally, a total of 22 hospitals including 49 departments (including department of dermatology, department of urology and department of

obstetrics and gynecology) were selected as study sites to include in this study. During the study period, the first 15 eligible individuals who arrived at each department every working day were invited to participate in the questionnaire survey and urine collection. The criteria for eligible participants were: (1) age $\geq$ 18 years; (2) having ever engaged in sexual activity; and (3) having not used any antibiotics in the last 2 weeks. The symptomatic infection was defined as the appearance of symptoms associated with gonorrhea or chlamydia infections, such as urethral discharge, vaginal discharge, dysuria or cervicitis. Ethical approval was provided by the Ethics Committee of Shenzhen Center for Chronic Disease Control (Approval No. 20180206). Written informed consent was obtained from all the participants.

## Data collection

The anonymous questionnaire was designed by the correspondent, with a total of 45 questions, and was conducted in Chinese only. Data were obtained on: socio-demographic characteristics (including age, gender, children, residency, local residence time, marital status, education, living arrangements status, insurance and occupation), sexual orientation, risky sexual behaviors, history of STI testing, history of STI infections, STI-related knowledge/attitude and self-reported symptoms related to STIs. After a preliminary questionnaire interview, a clinical examination was carried out for each patient. Information on symptoms (dysuria, painful urination, urethral discharge, etc.) was recorded. Subsequently, each eligible participant was invited to donate a urine specimen for CT and NG testing.

## Specimen collection and laboratory testing

15-30ml urine specimens were collected using the Cobas®urine specimen collection kit (Roche P/N 05170486190). These specimens were temporarily stored at 4˚C in local laboratories for up to 10 days before being transported to the central laboratory for testing. In the central laboratory, we used the MagNA Pure 96 System (Roche, Switzerland) to extract and purify DNA from urine specimens by an automated magnetic nucleic acid isolation method. Then, polymerase chain reaction (PCR) of the Cobas 4800® System (Roche, Switzerland) was performed using DNA extracted from urine specimens for testing CT and NG. Laboratory tests of CT and NG were performed based on standard procedures. Positive PCR results were confirmed as corresponding NG or CT infections.

## Statistical analyses

Considering the differences in symptomatic performance between men and women (especially with regard to asymptomatic infections), this study was stratified by gender. Descriptive analysis was conducted to describe frequencies and percentages of key variables, and to calculate the prevalence of asymptomatic gonorrhea or chlamydia. Statistical differences between asymptomatic NG or CT patients and non-patients for the categorical variables were assessed using Chi-square test and Fisher exact test as appropriate. Univariate logistic regressions were used to select appropriate variables for the multivariate logistic regression models. Those variables with $p$-value $< 0.2$ were included in the multivariate analyses to further examine the association between males/females with asymptomatic urethral NG and/or CT, and potential risk factors. Adjusted odds ratios (AORs) and their corresponding 95% confidence interval (CI) were calculated to measure the correlation strength. $P$-values $< 0.05$ were considered statistically significant. All analysis above were performed using SPSS 19.0.

**Table 1. Characteristics of all participants.**

| Variables | N (%) |
|---|---|
| Age, y | |
| ≤30 | 2934 (41.5) |
| >30 | 4136 (58.5) |
| Gender | |
| Male | 2258 (31.9) |
| female | 4812 (68.1) |
| Children | |
| No | 2679 (37.9) |
| Yes | 4338 (61.4) |
| Living arrangements status | |
| Living alone | 591 (8.4) |
| Living with spouse | 4716 (66.7) |
| Residence status | |
| Local residents | 1858 (26.3) |
| Migrants | 5212 (73.7) |
| Local residence time | |
| 0–12 months | 775 (11.0) |
| Over 1 year | 6295 (89.0) |
| Occupation | |
| Staff | 1803 (25.5) |
| Commercial services | 1529 (21.6) |
| Housework or unemployed | 1063 (15.0) |
| Worker | 1780 (25.2) |
| Highest educational level | |
| Lower than senior high school | 4268 (60.4) |
| Senior high school and above | 2802 (39.6) |
| Clinical settings | |
| Dermatological department | 1018 (14.4) |
| Gynecological department | 4136 (58.5) |
| Urological department | 1274 (18.0) |
| Family planning department | 624 (8.8) |
| Insurance | |
| Private/Medicaid | 4408 (62.3) |
| Uninsured | 2662 (37.7) |
| Sex with an anonymous partner in the last 3 months | |
| Yes | 2580 (36.5) |
| No | 4490 (63.5) |
| History of STI infections | |
| No | 5966 (84.4) |
| Yes | 1104 (15.6) |
| History of STI testing | |
| No | 6479 (91.6) |
| Yes | 591 (8.4) |
| STI-related knowledge | |
| Low | 5594 (79.1) |
| High | 1476 (20.9) |
| Partner notification | |
| No | 729 (10.3) |
| Yes | 6121 (86.6) |

## Results

### Prevalence and proportion of asymptomatic NG and/or CT infections

Between April 2018 and May 2018, 8,309 eligible individuals were invited to participate in this study. Of these, 7070 participants completed the questionnaire and provided urine specimens for molecular detection of NG and CT, so the survey response rate was 85.1%. The characteristics of all participants were shown in Table 1. In total, 182 (2.6%) participants were positive for NG and 648 (9.2%) for CT. The proportion of participants without symptoms was 2871/7070 (40.6%): 1120 (39.0%) for men and 1751 (61.0%) for women. By symptomatology, urogenital NG was detected in 17 of 2871 asymptomatic participants (0.6%), and urogenital CT was detected in 207 of 2871 asymptomatic participants (7.2%). Among men reporting no symptoms, the prevalence of NG and CT was 0.9% and 6.2%, respectively. Among women reporting no symptoms, the prevalence of NG and CT was 0.4% and 7.9%, respectively (Fig 1). The proportion of asymptomatic NG or CT infections was shown in Table 2.

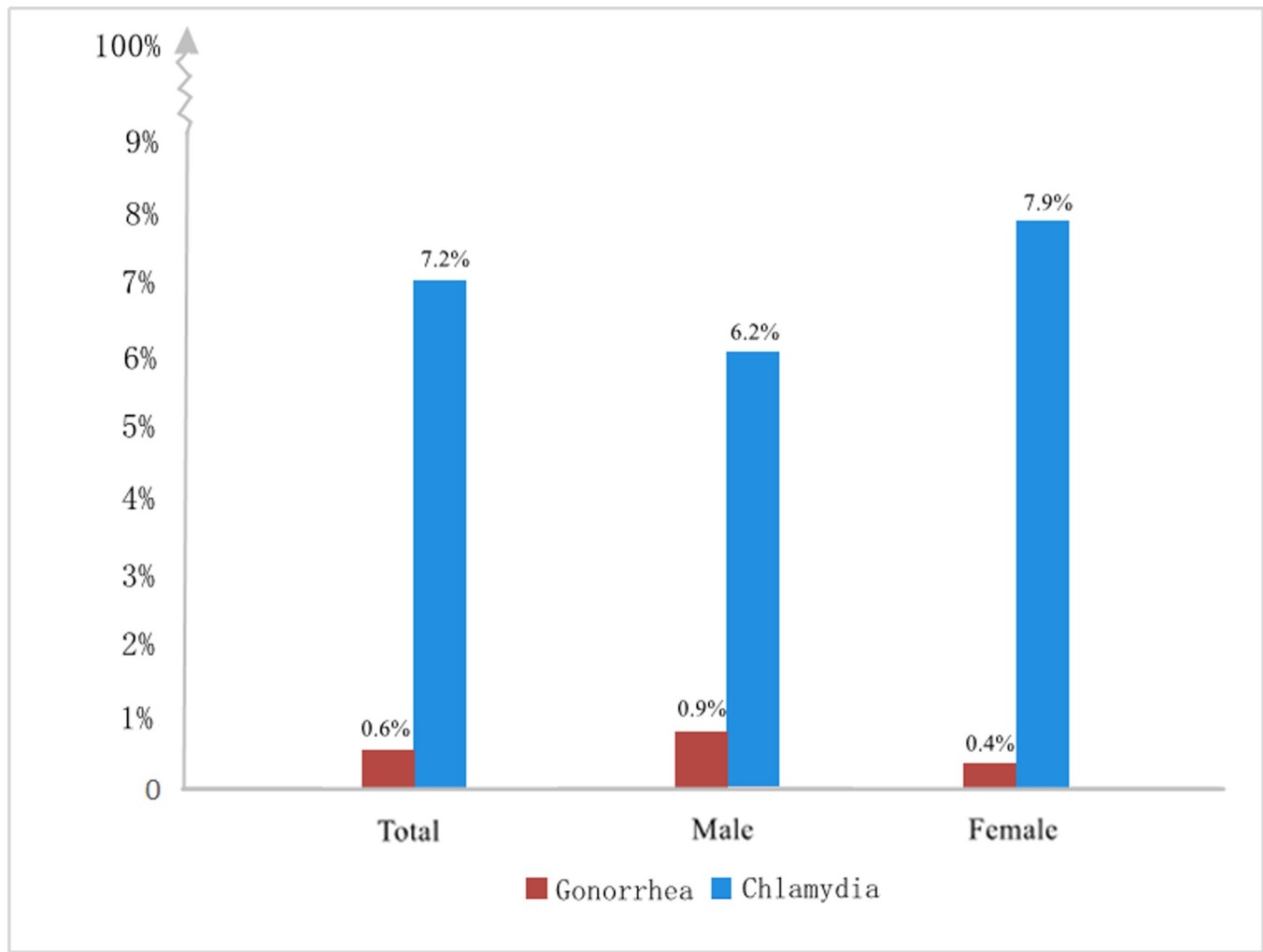

**Fig 1. Prevalence of asymptomatic Gonorrhea and Chlamydia infections by Gender.**

**Table 2. Gonorrhea and Chlamydia positive rate and proportion among 7070 STI clinic attenders stratified by sex and symptoms.**

| | Number tested (%) | Number +ve (%) | Gonorrhea | | Chlamydia | |
|---|---|---|---|---|---|---|
| | | | Symptomatic(+ve, %) | Asymptomatic(+ve, %) | Symptomatic(+ve, %) | Asymptomatic(+ve, %) |
| Male | 2258 (31.9) | 343 (15.2) | 129 (92.8%) | 10 (7.2%) | 175 (71.7%) | 69 (28.3%) |
| Female | 4812 (68.1) | 426 (8.9) | 36 (83.7%) | 7 (16.3%) | 266 (65.8%) | 138 (34.2%) |
| Total | 7070 (100) | 769 (10.9) | 165 (90.7%) | 17 (9.3%) | 441 (68.1%) | 207 (31.9%) |

### Characteristics of asymptomatic male participants

Of the 1120 asymptomatic male participants included in this analysis, 60.0% were over 30 of age; 73.7% were immigrants (unregistered residents of Shenzhen); 92.6% lived in Shenzhen for more than one year; 50.9% had education at the senior high school level or higher; 49.5% had had sex with an anonymous partner in the past 3 months; 87.5% had no history of STI infections; 91.2% had no history of STI testing and 77.6% had a low level of STI-related knowledge (Table 3).

Because of the low prevalence of NG in our study, we only described the results of factor analysis of asymptomatic CT infection. When compared with normal people, there were more asymptomatic men infected with CT in participants under 30 years of age (55.1% vs 39.0%), who were employed in commercial service work (39.1% vs 26.4%) and who were recruited through the urological department (66.7% vs 44.6%) (Table 3).

### Factors associated with asymptomatic CT infection among male participants

Risk factors associated with asymptomatic CT infection among male participants included males aged less than 30 years (aOR, 1.83; 95% CI, 1.11–3.03), being employed in commercial service work (2.82; 1.36–5.84) and being recruited through urological department (vs dermatological department, 2.12; 1.19–3.79) (Table 4).

### Characteristics of asymptomatic female participants

Of the1751 asymptomatic female participants included in this analysis, 59.0% were more than 30 years old;70.0% were immigrants (unregistered residents of Shenzhen); 90.6% lived in Shenzhen for more than one year; 70.5% had education at the senior high school level or higher; 26.4% had had sex with an anonymous partner in the last 3 months; 85.5% had no history of STI infections; 91.3% had no history of STI testing and 80.4% had a low level of STI-related knowledge (Table 5).

When compared with non-patients, there were more asymptomatic women infected with CT in participants under 30 years of age (55.8% vs 39.7%), who had no children (44.9% vs 33.9%), who were migrants (81.2% vs 69.1), and uninsured (42.8% vs 34.2%). In addition, most women with CT infection were employed in commercial service work (26.1% vs 18.6%), and had low STI-related knowledge (87.7% vs 79.7%) (Table 5).

### Factors associated with asymptomatic CT infection among female participants

Risk factors associated with asymptomatic CT infection among female participants included females aged less than 30 years (aOR, 1.78; 95% CI, 1.24–2.55) (Table 6).

**Table 3. Characteristics of asymptomatic male participants in Shenzhen, China.**

| Variables | | No. Total (%) N = 1120 | No. Negative cases (%) N = 1051 | No. Positive CT cases (%) N = 69 | P value |
|---|---|---|---|---|---|
| Age, y | ≤30 | 448 (40.0) | 410 (39.0) | 38 (55.1) | **0.008** |
| | >30 | 672 (60.0) | 641 (61.0) | 31 (44.9) | |
| Children | No | 606 (54.1) | 569 (54.1) | 37 (53.6) | 0.292 |
| | Yes | 510 (45.5) | 479 (45.6) | 31 (44.9) | |
| Living arrangements status | Living alone | 144 (12.9) | 138 (13.1) | 6 (8.7) | 0.079 |
| | Living with spouse | 651 (58.1) | 616 (58.6) | 35 (50.7) | |
| Residence status | Local residents | 295 (26.3) | 281 (26.7) | 14 (20.3) | 0.239 |
| | Migrants | 825 (73.7) | 770 (73.3) | 55 (79.7) | |
| Local residence time | 0–12 months | 83 (7.4) | 77 (7.3) | 6 (8.7) | 0.674 |
| | Over 1 year | 1037 (92.6) | 974 (92.7) | 63 (91.3) | |
| Occupation | Staff | 330 (29.5) | 319 (30.4) | 11 (15.9) | **0.010** |
| | Commercial services | 304 (27.1) | 277 (26.4) | 27 (39.1) | |
| | Housework or unemployed | 12 (1.1) | 11 (1.0) | 1 (1.4) | |
| | Worker | 334 (29.8) | 318 (30.3) | 16 (23.2) | |
| Highest educational level | Lower than senior high school | 550 (49.1) | 510 (48.5) | 40 (58.0) | 0.128 |
| | Senior high school and above | 570 (50.9) | 541 (51.5) | 29 (42.0) | |
| Clinical settings | Dermatological department | 362 (32.3) | 345 (32.8) | 17 (24.6) | **< 0.001** |
| | Urological department | 536 (47.9) | 490 (46.6) | 46 (66.7) | |
| | Family planning department | 220 (19.6) | 215 (20.5) | 5 (7.2) | |
| Insurance | Private/Medicaid | 724 (64.6) | 685 (65.2) | 39 (56.5) | 0.145 |
| | uninsured | 396 (35.4) | 366 (34.8) | 30 (43.5) | |
| Sex with an anonymous partner in the last 3 months | Yes | 554 (49.5) | 516 (49.1) | 38 (55.1) | 0.336 |
| | No | 566 (50.5) | 535 (50.9) | 31 (44.9) | |
| History of STI infections | No | 980 (87.5) | 920 (87.5) | 60 (87.0) | 0.790 |
| | Yes | 63 (5.7) | 58 (5.5) | 5 (7.2) | |
| History of STI testing | No | 1021 (91.2) | 956 (91.0) | 65 (94.2) | 0.358 |
| | Yes | 99 (8.8) | 95 (9.0) | 4 (5.8) | |
| STI-related knowledge | Low | 869 (77.6) | 817 (77.7) | 52 (75.4) | 0.647 |
| | High | 251 (22.4) | 234 (22.3) | 17 (24.6) | |
| Partner notification | No | 92 (8.2) | 86 (8.2) | 6 (8.7) | 0.493 |
| | Yes | 1007 (89.9) | 944 (89.8) | 63 (91.3) | |

## Discussion

In this clinic-based multi-site cross-sectional study, we determined both the prevalence and proportion of laboratory-confirmed urethral CT and/or NG infections that were asymptomatic among subjects presenting to clinics in Shenzhen, China. Overall, the prevalence of asymptomatic NG infection was low, but a high prevalence of CT infection was observed among males and females who have no symptoms. In addition, we found that about one-third of CT infections among males or females with urethral CT were asymptomatic. For asymptomatic males with urethral CT, we identified 3 independent predictors: (1) males under the

**Table 4. Factors associated with asymptomatic CT infections among male participants.**

| Variables | Univariate analysis | | Multivariate analysis | |
|---|---|---|---|---|
| | OR (95%CI) | *P* value | aOR (95%CI) | *P* value |
| Age, y | | | | |
| >30 | 1 | | **1** | |
| ≤30 | 1.92 (1.17–3.12) | 0.009 | **1.83 (1.11–3.03)** | **0.019** |
| Occupation | | | | |
| Staff | 1 | | 1 | |
| Commercial services | 2.83 (1.38–5.80) | 0.005 | **2.82 (1.36–5.84)** | **0.005** |
| Housework or unemployed | 2.64 (0.31–22.26) | 0.373 | – | – |
| Worker | 1.46 (0.67–3.19) | 0.344 | – | – |
| Highest educational level | | | | |
| Lower than senior high school | 1 | | 1 | |
| Senior high school and above | 0.68 (0.42–1.12) | 0.130 | 0.74 (0.42–1.30) | 0.291 |
| Clinical settings | | | | |
| Dermatological department | 1 | | 1 | |
| Urological department | 1.91 (1.07–3.38) | 0.027 | **2.12 (1.19–3.79)** | **0.011** |
| Family planning department | 0.47 (0.17–1.30) | 0.146 | 0.49 (0.18–1.36) | 0.172 |
| Insurance | | | | |
| Private/Medicaid | 1 | | 1 | |
| Uninsured | 1.44 (0.88–2.36) | 0.147 | 0.94 (0.54–1.66) | 0.842 |

age of 30; (2) being employed in the commercial service work; and (3) being recruited through the urological department (vs dermatological department). For asymptomatic females with urethral CT, age less than 30 years was a risk factor.

The overall prevalence of CT and NG observed in this study was lower than that of Zhang et al[6] in 2009 in the similar population in Shenzhen but still significantly higher than the national level.[7] Currently, the cause for the decline in the prevalence of NG and CT are not clear, but these two pathogens (especially CT) are still serious public health problems in Shenzhen. A systematic review suggested that if the prevalence of CT is between 3.1–10.0%, the screening for CT infectious is cost-effective.[13] Our findings indicated that a regular and comprehensive CT screening is warranted in Shenzhen.

The proportion of asymptomatic NG and CT has been widely reported, but results vary widely around the world, ranging from 8% to 87%.[14–17] The disparities in the proportion of asymptomatic participants with or without STIs may be attributed to the different laboratory methods employed, the distribution of risk factors and the composition of the population studied. Given that one-third of CT infections are asymptomatic, passive screening in Shenzhen (i.e., screening patients for medical treatment) is not enough. We need to identify risk factors for patients with asymptomatic CT infections and conduct targeted screening across the entire population.

The mechanism for why some people infected with CT or NG are asymptomatic remains uncertain; it is possibly related to a low-level bacterial load.[18] Our epidemiological study has shown that participants (both males and females) under the age of 30 are associated with higher odds of a asymptomatic STI positive screening result. This is similar to the findings from other previous studies which investigated risk factors for the overall prevalence of NG or CT.[19–23] Some potential explanations for the impact of age on the prevalence of STI might be: first, youth are often accompanied by strong sexual desire, which results in frequent sexual activity; second, because of a lack of awareness of sexual safety, youth are more likely to engage

**Table 5. Characteristics of asymptomatic female participants.**

| Variables | | No. Total (%)<br>N = 1751 | No. Negative cases (%)<br>N = 1613 | No. Positive CT cases (%)<br>N = 138 | P value |
|---|---|---|---|---|---|
| Age, y | ≤30 | 718 (41.0) | 641 (39.7) | 77 (55.8) | < **0.001** |
| | >30 | 1033 (59.0) | 972 (60.3) | 61 (44.2) | |
| Children | NO | 609 (34.8) | 547 (33.9) | 62 (44.9) | **0.019** |
| | Yes | 1128 (64.4) | 1054 (65.3) | 74 (53.6) | |
| Living arrangements status | Living alone | 114 (6.5) | 104 (6.4) | 10 (7.2) | 0.220 |
| | Living with spouse | 1325 (75.7) | 1247 (77.3) | 78 (56.5) | |
| Residence status | Local residents | 525 (30.0) | 499 (30.9) | 26 (18.8) | **0.003** |
| | Migrants | 1226 (70.0) | 1114 (69.1) | 112 (81.2) | |
| Local residence time | 0–12 months | 165 (9.4) | 147 (9.1) | 18 (13.0) | 0.130 |
| | Over 1 year | 1586 (90.6) | 1466 (90.9) | 120 (87.0) | |
| Occupation | Staff | 492 (28.1) | 460 (28.5) | 32 (23.2) | **0.017** |
| | Commercial services | 336 (19.2) | 300 (18.6) | 36 (26.1) | |
| | Housework or unemployed | 375 (21.4) | 351 (21.8) | 24 (17.4) | |
| | Worker | 279 (15.9) | 248 (15.4) | 31 (22.5) | |
| Highest educational level | Lower than senior high school | 516 (29.5) | 465 (28.9) | 53 (38.4) | 0.320 |
| | Senior high school and above | 1235 (70.5) | 1144 (71.1) | 85 (61.6) | |
| Clinical settings | dermatological department | 75 (4.3) | 65 (4.0) | 10 (7.2) | 0.260 |
| | gynecological department | 1367 (78.1) | 1259 (78.1) | 108 (78.3) | |
| | Family planning department | 276 (15.8) | 259 (16.1) | 17 (12.3) | |
| Insurance | Private/Medicaid | 1140 (65.1) | 1061 (65.8) | 79 (57.2) | **0.044** |
| | Uninsured | 611 (34.9) | 552 (34.2) | 59 (42.8) | |
| Sex with an anonymous partner in the last 3 months | Yes | 463 (26.4) | 423 (26.2) | 40 (29.0) | 0.482 |
| | No | 1288 (73.6) | 1190 (73.8) | 98 (71.0) | |
| History of STI infections | No | 1497 (85.5) | 1377 (85.4) | 120 (87.0) | 0.862 |
| | Yes | 76 (4.3) | 71 (4.4) | 5 (3.6) | |
| History of STI testing | No | 1599 (91.3) | 1467 (90.9) | 132 (95.7) | 0.059 |
| | Yes | 152 (8.7) | 146 (9.1) | 6 (4.3) | |
| STI-related knowledge | Low | 1407 (80.4) | 1286 (79.7) | 121 (87.7) | **0.025** |
| | High | 344 (19.6) | 327 (20.3) | 17 (12.3) | |
| Partner notification | No | 88 (5.0) | 79 (4.9) | 9 (6.5) | 0.396 |
| | Yes | 1594 (91.0) | 1468 (91.0) | 126 (91.3) | |

in risky sexual behaviors. In view of this, some developed countries have launched STI screening programs for young adults.[20, 24, 25] Our findings suggest that attention should also be paid to asymptomatic STIs during the screening process.

Another noteworthy finding was that the prevalence of asymptomatic urogenital CT was significantly higher in male participants who were recruited through the urological department than that of male participants who were recruited through the dermatological department. In the past, STI screening services conducted by local government were always carried out with the dermatological department as the core place. However, our finding indicates that

**Table 6. Factors associated with asymptomatic CT infections among female participants.**

| Variables | Univariate analysis | | Multivariate analysis | |
|---|---|---|---|---|
| | OR (95%CI) | *P* value | aOR (95%CI) | *P* value |
| Age, y | | | | |
| >30 | 1 | | **1** | |
| ≤30 | 1.96 (1.39–2.78) | 0.001 | **1.78 (1.24–2.55)** | **0.002** |
| Children | | | | |
| NO | 1 | | 1 | |
| Yes | 0.64 (0.45–0.90) | 0.011 | 0.74 (0.48–1.15) | 0.177 |
| Residence status | | | | |
| Local residents | 1 | | 1 | |
| Migrants | 1.91 (1.24–2.94) | 0.003 | 1.52 (0.95–2.43) | 0.079 |
| Local residence time | | | | |
| 0–12 months | 1 | | 1 | |
| Over 1 year | 0.83 (0.64–1.08) | 0.168 | 0.91 (0.69–1.19) | 0.487 |
| Occupation | | | | |
| Staff | 1 | | | |
| Commercial services | 1.78 (1.08–2.92) | 0.023 | 1.46 (0.86–2.46) | 0.160 |
| Housework or unemployed | 1.07 (0.63–1.83) | 0.802 | – | – |
| Worker | 1.86 (1.11–3.11) | 0.018 | 1.51 (0.87–2.63) | 0.142 |
| Clinical settings | | | | |
| Gynecological department | 1 | | 1 | |
| Dermatological department | 1.79 (0.90–3.59) | 0.099 | 2.02 (0.98–4.14) | 0.056 |
| Family planning department | 0.77 (0.45–1.30) | 0.321 | – | – |
| Insurance | | | | |
| Private/Medicaid | 1 | | 1 | |
| Uninsured | 1.44 (1.01–2.04) | 0.044 | 1.05 (0.70–1.57) | 0.809 |
| History of STI test | | | | |
| No | 1 | | 1 | |
| Yes | 0.46 (0.20–1.05) | 0.066 | 0.56 (0.24–1.32) | 0.182 |
| STI-related knowledge | | | | |
| Low | 1 | | 1 | |
| High | 0.55 (0.33–0.93) | 0.026 | 0.62 (0.36–1.08) | 0.092 |

urological department may be a better site to offer male patients opportunistic screening for CT. So far, although most developed countries have implemented opportunistic screening services for STIs, the effectiveness of these services in reducing the prevalence of STIs has not been satisfactory.[20, 26, 27] Our results may provide room for improvement of STIs screening. In addition to the screening site, low uptake rates of STIs screening may also be one of the factors hindering the screening effectiveness. According to a study from the US, approximately 37.9% females reported ever receiving a CT testing.[28] In China, Wu et al[29] found that less than one-third of males had participated in a NG or CT testing. Surprisingly, we observed that the screening intention was significantly lower in male participants who were recruited through the dermatological department than that of male participants who were recruited through the urological department (S1 Fig). People with a higher willingness to screen are more likely to go to the urological department, which further supports the placement of screening sites for males in the urological department.

Our study has some limitations. First, our study was limited to clinics where participants who had urethral symptoms and who were at particularly high-risk of infection were more

likely to go to the clinics for treatment. It is possible that the prevalence of NG and/or CT infections among participants without symptoms in the community will be overestimated. Thus, caution should be used when generalizing our findings to the community in Shenzhen. Second, the participants of the study were recruited using a convenient sampling method and the sampling period was only 1 month, which may also lead to a potential selection bias. Third, we limited the detection to urogenital specimens only, so extragenital infections were not captured. However, given that the high-risk population with rectal or oropharyngeal infections (such as MSM and sex workers) included in our study was rare, our data on prevalence of NG and/or CT infection were reliable. Finally, as with other cross-sectional studies, reporting bias, recall bias and limitation in making causal inferences should be considered.

In conclusion, a substantial prevalence of asymptomatic CT infection was found among males and females presenting to clinics in Shenzhen. The significant correlation between asymptomatic CT infection and age as well as clinical setting could help identify high-risk populations and guide resource allocation and screening. Future studies should investigate people's willingness to screen and the effectiveness of screening strategies on these high-risk populations.

## Supporting information

**S1 Fig. Impact of different clinical settings on screening intention.**
(TIFF)

**S1 Questionnaire.**
(DOC)

**S1 Data.**
(SAV)

## Acknowledgments

The authors thank all the staff in Shenzhen Center for Chronic Disease Control who made many contributions to this study.

## Author Contributions

**Conceptualization:** Kang-Kang Chen, Yu-Mao Cai.

**Data curation:** Shu-Xia Chang, Xiao-Ting Liu.

**Investigation:** Nan Xia, Pei-Sheng Xiong.

**Writing – original draft:** Shu-Xia Chang, Kang-Kang Chen.

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
