## [Decision Letter · Decision Letter 0]

20 Apr 2020

PONE-D-20-07301

Prevalence and risk factors of asymptomatic Neisseria gonorrhoeae and Chlamydia trachomatis infections in Shenzhen, China: a cross-sectional study.

PLOS ONE

Dear Mr. Cai,

Thank you for submitting your manuscript to PLOS ONE. After careful consideration, we feel that it has merit but does not fully meet PLOS ONE’s publication criteria as it currently stands. Therefore, we invite you to submit a revised version of the manuscript that addresses the points raised during the review process.

We would appreciate receiving your revised manuscript by Jun 04 2020 11:59PM. To enhance the reproducibility of your results, we recommend that if applicable you deposit your laboratory protocols in protocols.io, where a protocol can be assigned its own identifier (DOI) such that it can be cited independently in the future. For instructions see: http://journals.plos.org/plosone/s/submission-guidelines#loc-laboratory-protocols

We look forward to receiving your revised manuscript.

Kind regards,

Remco PH Peters, MD, PhD, DLSHTM

Academic Editor

PLOS ONE

2. Please include additional information regarding the survey or questionnaire used in the study and ensure that you have provided sufficient details that others could replicate the analyses. If you developed and/or translated a questionnaire as part of this study and it is not under a copyright license more restrictive than Creative Commons Attribution (CC-BY), please include a copy, in both the original language and English, as Supporting Information.

Reviewers' comments:

Reviewer's Responses to Questions

**Comments to the Author**

1. Is the manuscript technically sound, and do the data support the conclusions?

Reviewer #1: Partly

Reviewer #2: Yes

2. Has the statistical analysis been performed appropriately and rigorously? 

Reviewer #1: Yes

Reviewer #2: Yes

3. Have the authors made all data underlying the findings in their manuscript fully available?

Reviewer #1: No

Reviewer #2: No

4. Is the manuscript presented in an intelligible fashion and written in standard English?

Reviewer #1: No

Reviewer #2: Yes

5. Review Comments to the Author

Reviewer #1: Thank you for the opportunity to review your interesting and important research

Major comments

1. The aim of the study was to determine prevalence of asymptomatic Chlamydia trachomatis and Neisseria gonorrhoeae infections and the risk factors associated with these infections. However, as the authors have already conceded that due to low a prevalence of N. gonorrhoeae in their study, the number of C. trachomatis and N. gonorrhoeae cases were combined therefore the risks factors associated with asymptomatic infection of each pathogen could not be independently determined. Could you discuss in your limitations the implications of this? I think the title of the study in that regard does not address the aims as the risk factors associated with asymptomatic C. trachomatis and N. gonorrhoeae were not independently determined in this study.

2. The authors determined one the risk factors for asymptomatic C. trachomatis / N. gonorrhoeae was recruitment from urology department (men) and dermatology (men). Did you include results from the other 46 departments in your analysis? In your results in Table 3 and 5 data from only three departments is shown.

3. Results in table 2 and table 3 should also include the actual number of positive C. trachomatis and N. gonorrhoeae cases not just as “positive cases”

Minor comments

Line 42 and 80: “Chlamydia trachomatis” and “Neisseria gonorrhoeae” should always be in italics

Line 82: can you cite a more recent reference with up-to-date epidemiological information

Line 86: please add a reference to support mentioned statements

Line 150: “magnetism” should be “magnetic”

Line 264: “but these two diseases (especially CT)” please rephrase as CT and NG are bacterial names not diseases

Results: were there any participants with dual C. trachomatis or N. gonorrhoeae infections?

Reviewer #2: Asymptomatic infection is the main source of STIs. The results of this study are interesting for readers to understand the burden and characteristics of asymptomatic gonorrhea and chlamydia infection in Shenzhen, as well as for the formulating corresponding control strategies. In general, this is a good-writing manuscript with interesting findings.

The following suggestions for your considering in your revision.

１. Please clarify the selection process: Are these 6 districts reported more cases than the 4 who were not selected?(Line 121: First, we selected 6 out of the 10 administrative districts in Shenzhen based on the number of NG and CT cases reported in 2017)．

２. Please clarify the enrollment: (Line 127)The first 15 eligible individuals who arrived at each department were invited…… ), this is confusing, I guess should be “the first 15 eligible individual every working day” to participate in the questionnaire survey and urine collection

３. Definition of symptoms, as this is quite important and easy to cause bias. From the results, it seemed that symptomatic or asymptomatic were self-reported, the question is who made the judgement? Is any case or all cases examined by doctors? As patients may reported symptom-free, while doctor found urethral discharge during physical examination. In addition, urethritis or cervicitis are clinical diagnosis, and should be judged by clinical examination, please clarify this.

４. Please delete the repeated sentence: Line 181: The proportion of asymptomatic urethral NG in men with urethral NG was 7.2%; for women it was 16.3%. The proportion of asymptomatic

５. In the results, a table which describe the characteristics of all participants will add more information to readers.

６. Table 1 is confusing, will be clearer to have: number of total tested, number of positive in total; number of asymptomatic, number of positive among asymptomatic, this could be further stratified by sex.

6. PLOS authors have the option to publish the peer review history of their article (what does this mean?). If published, this will include your full peer review and any attached files.

Reviewer #1: No

Reviewer #2: None

---

## [Author Response · Author response to Decision Letter 0]

19 May 2020

Dear Editors and Reviewers:

Thank you for your letter and for the reviewers’ comments concerning our manuscript entitled “Prevalence and risk factors of asymptomatic Neisseria gonorrhoeae and Chlamydia trachomatis infections in Shenzhen, China: a cross-sectional study.” (ID: PONE-D-20-07301). Those comments are all valuable and very helpful for revising and improving our paper, as well as the important guiding significance to our researches. We have studied comments carefully and have made correction which we hope meet with approval. The main corrections in the paper and the responds to the reviewer’s comments are as following:

1. Please ensure that your manuscript meets PLOS ONE's style requirements, including those for file naming. The PLOS ONE style templates can be found at http://www.plosone.org/attachments/PLOSOne_formatting_sample_main_body.pdf. http://www.plosone.org/attachments/PLOSOne_formatting_sample_title_authors_affiliations.pdf.

Response: we can’t get the files in the above two links (404 Not Found). We can only adjust the style of the manuscript according to the “instructions for authors” on the website.

2. Please include additional information regarding the survey or questionnaire used in the study and ensure that you have provided sufficient details that others could replicate the analyses. If you developed and/or translated a questionnaire as part of this study and it is not under a copyright license more restrictive than Creative Commons Attribution (CC-BY), please include a copy, in both the original language and English, as Supporting Information.

Response: questionnaire has been uploaded as supporting information.

3. If there are no restrictions, please upload the minimal anonymized data set necessary to replicate your study findings as either Supporting Information files or to a stable, public repository and provide us with the relevant URLs, DOIs, or accession numbers. Please see http://www.bmj.com/content/340/bmj.c181.long for guidelines on how to de-identify and prepare clinical data for publication. For a list of acceptable repositories, please see http://journals.plos.org/plosone/s/data-availability#loc-recommended-repositories.

Response: data set has been uploaded as supporting information.

Response: corresponding author’s account has been linked to ORCID iD.

Responds to the reviewer#1’s comments:

Major comments:

1. Response to comment: The aim of the study was to determine prevalence of asymptomatic Chlamydia trachomatis and Neisseria gonorrhoeae infections and the risk factors associated with these infections. However, as the authors have already conceded that due to low a prevalence of N. gonorrhoeae in their study, the number of C. trachomatis and N. gonorrhoeae cases were combined therefore the risks factors associated with asymptomatic infection of each pathogen could not be independently determined. Could you discuss in your limitations the implications of this? I think the title of the study in that regard does not address the aims as the risk factors associated with asymptomatic C. trachomatis and N. gonorrhoeae were not independently determined in this study.

Response: Thank you for your comment. After careful consideration of the reviewer’s comment, we think the reviewer’s suggestion is very reasonable. Therefore, we re-analyzed the risk factors of asymptomatic infections of NG and CT. The detailed process is shown below.

Table R1, Table R2 and Table R3 are the analysis of risk factors for asymptomatic NG&CT, CT and NG infections in men, respectively. After comparing Table R1 and Table R2, we found that the identified risk factors did not change (the OR and 95% CI changed slightly). After analyzing the risk factors of asymptomatic NG infection (Table R3), we found that living arrangements status was a predictor. However, among the male participants in Shenzhen, there were only 10 asymptomatic NG cases, and the prevalence of asymptomatic NG infection was only 0.9%. Due to low prevalence, the upper and lower limits of the 95%CI of living arrangements status fluctuate greatly (0.01-0.37), resulting in reduced accuracy. Considering that asymptomatic NG infection among men is not a serious public health problem in Shenzhen, we think it is unnecessary to do factor analysis for NG infection among male participants in our study.

Table R4, Table R5 and Table R6 are the analysis of risk factors for asymptomatic NG&CT, CT and NG infections in women, respectively. After comparing Table R4 and Table R5, we found that the variable (clinical settings) was no longer a predictor and the OR value for age changed slightly. After analyzing the risk factors of asymptomatic NG infection (Table R6), we found that the clinical setting was a predictor. Similarly, we found that the upper and lower limits of the 95%CI fluctuate greatly (3.39-74.46). In addition, among female participants in Shenzhen, there were only 7 asymptomatic NG cases, and the prevalence of asymptomatic NG infection was only 0.4%. Therefore, we also think it is unnecessary to do factor analysis for NG infection among female participants in our study.

According to the above analysis, we think it is necessary to retain the prevalence and proportion of NG infection, but factor analysis for NG infection is not required. The original research purpose “The aims of this study were to investigate the prevalence and proportion of laboratory-confirmed urethral Chlamydia trachomatis (CT) and Neisseria gonorrhoeae (NG) infections that were asymptomatic among individuals presenting to clinics in Shenzhen and the risk factors related to these asymptomatic infections.” has been changed to “The aims of this study were to investigate the prevalence and proportion of laboratory-confirmed urethral Chlamydia trachomatis (CT) and Neisseria gonorrhoeae (NG) infections that were asymptomatic among individuals presenting to clinics in Shenzhen and the risk factors related to asymptomatic CT infection.”. Please see title, abstract (line 43) and line 111. In addition, in the results (risk analysis section), we only describe the results of factor analysis of asymptomatic CT infection. Please see line 199-line 204, line 207-line 211, line 222-line 227 and line 231-line 233.

2. Response to comment: The authors determined one the risk factors for asymptomatic C. trachomatis / N. gonorrhoeae was recruitment from urology department (men) and dermatology (men). Did you include results from the other 46 departments in your analysis? In your results in Table 3 and 5 data from only three departments is shown.

Response: Thank you for your comment. This study mainly took STIs-related clinics as the study sites, because the individuals seeking medical services in these clinical settings are usually a high-risk group of STIs. This group has a heavy burden of STIs, which is sufficient to be considered a cost-effective intervention target for NG and CT infections. Therefore, the 49 departments in our study are all composed of dermatological department, gynecological department, urological department and family planning department. Our study included 22 hospitals. Among them, many are maternal and child health hospitals or centers for chronic disease control. These hospitals usually do not have urological department and family planning department. Therefore, 22 hospital (49 departments) were eventually included in our study.

3. Response to comment: Results in table 2 and table 3 should also include the actual number of positive C. trachomatis and N. gonorrhoeae cases not just as “positive cases”

Response: “positive cases” has been modified to “positive CT cases”. Please refer to the reply to comment 1 for the reason for modification.

Minor comments:

Response to comment: Line 42 and 80: “Chlamydia trachomatis” and “Neisseria gonorrhoeae” should always be in italics

Line 82: can you cite a more recent reference with up-to-date epidemiological information

Line 86: please add a reference to support mentioned statements

Line 150: “magnetism” should be “magnetic”

Line 264: “but these two diseases (especially CT)” please rephrase as CT and NG are bacterial names not diseases

Results: were there any participants with dual C. trachomatis or N. gonorrhoeae infections?

Response: We have made correction according to the reviewer’s comment. Please see line 41, line 77, line 79, line 83, line 151 and line 252.

Among the male participants, there were 10 cases of dual infection of NG and CT, and the prevalence was 0.4%. Among the female participants, there were 3 cases of dual infection of NG and CT, and the prevalence was 0.2%. Due to the low prevalence, we did not describe them in the results.

Responds to the reviewer#2’s comments:

1. Response to comment: Please clarify the selection process: Are these 6 districts reported more cases than the 4 who were not selected? (Line 121: First, we selected 6 out of the 10 administrative districts in Shenzhen based on the number of NG and CT cases reported in 2017).

Response: Thank you for your comment. We have made correction according to the reviewer’s comment. Please see page 5, line 118- line 121.

2. Response to comment: Please clarify the enrollment: (Line 127) The first 15 eligible individuals who arrived at each department were invited…), this is confusing, I guess should be “the first 15 eligible individual every working day” to participate in the questionnaire survey and urine collection

Response: Thank you for your comment. We have made correction according to the reviewer’s comment. Please see page 5, Line 126.

3. Response to comment: Definition of symptoms, as this is quite important and easy to cause bias. From the results, it seemed that symptomatic or asymptomatic were self-reported, the question is who made the judgement? Is any case or all cases examined by doctors? As patients may reported symptom-free, while doctor found urethral discharge during physical examination. In addition, urethritis or cervicitis are clinical diagnosis, and should be judged by clinical examination, please clarify this.

Response: Thank you for your comment. When the clinic attenders agreed to participate in the questionnaire, we first asked them to describe whether they felt uncomfortable. Then, regardless of whether the participants reported symptoms or not, a clinician would do a physical examination for the respondent and recorded the corresponding results. The definition of symptoms in this study was based on the doctor’s examination results, so the results are reliable. In order to allow readers to understand our research more clearly, we have added this content in the method section, please see line 140-line 143.

4. Response to comment: Please delete the repeated sentence: Line 181: The proportion of asymptomatic urethral NG in men with urethral NG was 7.2%; for women it was 16.3%. The proportion of asymptomatic

Response: Thank you for your comment. The sentences “The proportion of asymptomatic urethral NG in men with urethral NG was 7.2%; for women it was 16.3%. The proportion of asymptomatic urethral CT in men with urethral CT was 28.3%; for women it was 34.2%.” have been deleted.

5. Response to comment: In the results, a table which describe the characteristics of all participants will add more information to readers.

Response: Thanks to the Reviewer for this comment. The characteristics of all participants were shown in table 1.

6. Response to comment: Table 1 is confusing, will be clearer to have: number of total tested, number of positive in total; number of asymptomatic, number of positive among asymptomatic, this could be further stratified by sex.

Response: We have made correction according to the Reviewer’s comment. Please see table 2.

We tried our best to improve the manuscript and made some changes in the manuscript. These changes will not influence the content and framework of the paper. And here we did not list the changes but marked in red in revised paper. We appreciate for Editors/Reviewers’ warm work earnestly and hope that the correction will meet with approval. Once again, thank you very much for your comments and suggestions.

---

## [Editor Report · Decision Letter 1]

22 May 2020

Cross-sectional study of asymptomatic Neisseria gonorrhoeae and Chlamydia trachomatis infections in sexually transmitted disease related clinics in Shenzhen, China

PONE-D-20-07301R1

Dear Dr. Cai,

We are pleased to inform you that your manuscript has been judged scientifically suitable for publication and will be formally accepted for publication once it complies with all outstanding technical requirements.

With kind regards,

Remco PH Peters, MD, PhD, DLSHTM

Academic Editor

PLOS ONE
---

## [Editor Report · Acceptance letter]

29 May 2020

PONE-D-20-07301R1 

 Cross-sectional study of asymptomatic Neisseria gonorrhoeae and Chlamydia trachomatis infections in sexually transmitted disease related clinics in Shenzhen, China 

Dear Dr. Cai:

I am pleased to inform you that your manuscript has been deemed suitable for publication in PLOS ONE. Congratulations! Your manuscript is now with our production department. 

With kind regards,

on behalf of

Prof Remco PH Peters 

Academic Editor

PLOS ONE